# Advancing Spiking Neural Networks for Sequential Modeling with Central Pattern Generators

**Changze Lv**[1*]   **Dongqi Han**[2†]   **Yansen Wang**[2†]   **Xiaoqing Zheng**[1†]
**Xuanjing Huang**[1]   **Dongsheng Li**[2]
[1]School of Computer Science, Fudan University
[2]Microsoft Research Asia
{czlv22}@m.fudan.edu.cn, {zhengxq,xjhuang}@fudan.edu.cn,
{yansenwang,dongqihan,dongsli}@microsoft.com

## Abstract

Spiking neural networks (SNNs) represent a promising approach to developing artificial neural networks that are both energy-efficient and biologically plausible. However, applying SNNs to sequential tasks, such as text classification and time-series forecasting, has been hindered by the challenge of creating an effective and hardware-friendly spike-form positional encoding (PE) strategy. Drawing inspiration from the central pattern generators (CPGs) in the human brain, which produce rhythmic patterned outputs without requiring rhythmic inputs, we propose a novel PE technique for SNNs, termed CPG-PE. We demonstrate that the commonly used sinusoidal PE is mathematically a specific solution to the membrane potential dynamics of a particular CPG. Moreover, extensive experiments across various domains, including time-series forecasting, natural language processing, and image classification, show that SNNs with CPG-PE outperform their conventional counterparts. Additionally, we perform analysis experiments to elucidate the mechanism through which SNNs encode positional information and to explore the function of CPGs in the human brain. This investigation may offer valuable insights into the fundamental principles of neural computation. Our code is available at https://github.com/microsoft/SeqSNN.

## 1   Introduction

Spiking neural network (SNN) [1] has increasingly attracted research interests in recent years, primarily due to its energy efficiency, event-driven paradigm, biological plausibility, and other distinctive properties. The spiking neurons in SNN are dynamical systems that generate binary signals (spike or non-spike) and communicate these signals like artificial neural networks (ANNs) for computation [2–9]. Many advanced architectures and methodologies developed for ANNs are also applicable to SNNs, enhancing their capabilities. Notable among these are backpropagation [10], batch normalization [11, 12], and Transformer architecture [4, 13, 5, 6], which collectively broaden the functional scope of SNNs.

Despite the promising advances in SNNs, several challenges persist when adapting them to diverse tasks. A fundamental challenge is that SNNs, which are event-triggered, lack robust and effective mechanisms to capture indexing information, rhythmic patterns, and periodic data. This limitation can adversely affect SNNs' ability to process and analyze different data modalities, including natural language, and time series. Meanwhile, while SNNs aim to emulate the neural circuits of the brain,

---

[*]The work was conducted during the internship of Changze Lv at Microsoft Research Asia.
[†]Corresponding authors.

38th Conference on Neural Information Processing Systems (NeurIPS 2024).

their reliance on spike-based communication imposes limitations. Consequently, not all deep learning techniques applicable to ANNs can be directly transferred to SNNs. For instance, methods like HiPPO [14] or trigonometric positional encoding [15] are not readily compatible with the spike format used in SNNs. Moreover, even the most state-of-the-art ANNs still lag significantly behind human capabilities in many tasks [16, 17]. Therefore, to enhance the functionality of SNNs, one promising approach is to draw further inspiration from biological neural mechanisms. In this regard, we propose the analogy of central pattern generators (CPGs) [18], a kind of neural circuit well-known in neuroscience, with positional encoding (PE), a technique extensively utilized in deep learning. This analogy is designed to operate within the SNN framework, potentially bridging the gap between biologically inspired models and modern deep learning techniques.

In neuroscience, a CPG (See Figure 2 for an illustration) is a group of neurons capable of producing rhythmic patterned outputs without requiring rhythmic inputs [19, 20]. These neural circuits are found in the spinal cord and brainstem and are responsible for generating the rhythmic signals that control vital activities such as locomotion, respiration, and chewing [21].

On the other hand, PE is an important technique for ANNs, particularly within models tailored for sequence processing task [15, 22, 23]. By endowing each element of the input sequence with positional information, typically achieved through diverse mathematical formulations or learnable embeddings, neural networks acquire the capability to discern the order and relative positions of the elements within the sequence.

We argue that these two concepts, despite seemingly unrelated, can be connected profoundly. Intuitively, CPG and PE both generate periodic outputs (with respect to time for CPG and with respective to position for PE). Moreover, in this paper, we reveal a deeper relationship between these two concepts by showing that **the widely used sinusoidal PE is mathematically a particular solution of the membrane potential dynamics of a specific CPG**.

However, current SNN architectures exhibit a notable deficiency in implementing an effective and biologically plausible PE mechanism. Existing so-called positional encoding methods for SNNs [4, 5] rely on input data, often resulting in non-spike and repetitive outputs for different positions. Furthermore, incorporating PE techniques designed for ANNs necessitates the calculation of membrane potentials, which is incompatible with the spike format of SNNs. To address these issues, we draw inspiration from the spiking properties of the CPGs and propose a straightforward yet versatile PE technique for SNNs, termed CPG-PE. This method encodes positional information with multiple neurons with various patterns of spike trains. To summarize the highlights of our study:

- **Novel Positional Encoding for SNNs.** We introduce a bio-plausible and effective PE approach tailored specifically for SNNs. This innovative strategy draws inspiration from the central pattern generator found in the human brain. Additionally, we propose a straightforward implementation of CPG-PE in SNNs, which is also compatible with neuromorphic hardware as it can be realized using circuits of leaky integrate-and-fire neurons.

- **Consistent Performance Gain.** Our proposed methods significantly and consistently enhance the performance of SNNs across a wide range of sequential tasks, including time-series forecasting and text classification.

- **Insightful Analysis.** Our research represents one of the pioneering efforts to comprehensively analyze (1) the mechanism by which SNNs capture positional information and (2) the role of CPGs in the brain. This analysis provides valuable insights into the underlying principles of neural computation.

## 2 Preliminaries

### 2.1 Spiking Neural Networks

The basic unit in SNNs is the spiking neuron, such as the leaky integrate-and-fire (LIF) neuron [1], which operates based on an input current $I(t)$ and influences the membrane potential $U(t)$ and the

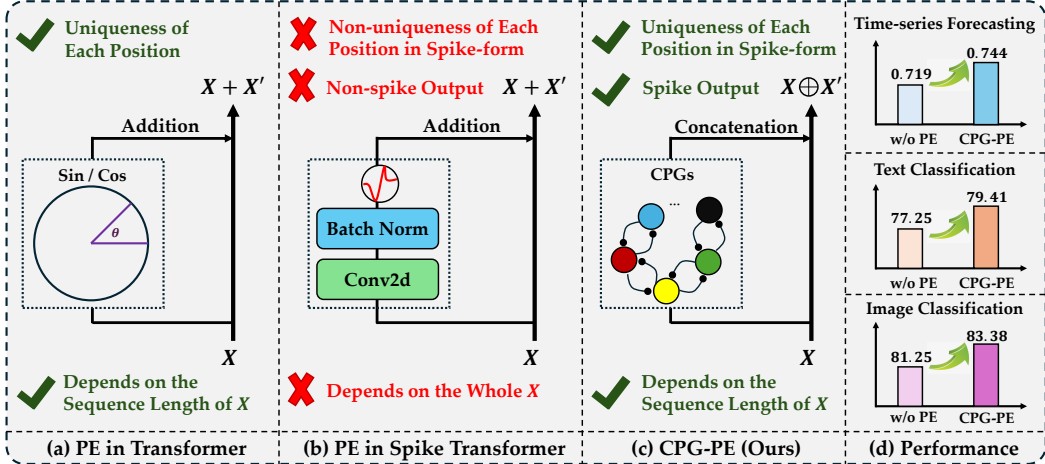

Figure 1: (a) Positional encoding (PE) in ANN Transformers. (b) Relative PE [‡] in Spike Transformers [4–6]. (c) Our Proposed CPG-PE method. (d) CPG-PE consistently improves learning performance across various tasks. CPG-PE is an ideal PE method tailored for SNNs, detailed in Section 3.

spike $S(t)$ at time $t$. The dynamics of the LIF neuron are described by the following equations:

$$U(t) = H(t - \Delta t) + I(t), \quad I(t) = f(\mathbf{x}; \theta), \tag{1}$$

$$H(t) = V_{reset} S(t) + (1 - S(t)) \beta U(t), \tag{2}$$

$$S(t) = \begin{cases} 1, & \text{if } U(t) \geq U_{\text{thr}} \\ 0, & \text{if } U(t) < U_{\text{thr}} \end{cases}. \tag{3}$$

Here, $I(t)$ is the spatial input to the LIF neuron at time step $t$, calculated using the function $f$ with $\mathbf{x}$ as input and $\theta$ as learnable parameters. $\Delta t$ is the discretization constant that determines the granularity of LIF modeling, and $H(t)$ is the temporal output of the neuron at time step $t$. The spike $S(t)$ is defined as a Heaviside step function based on the membrane potential. When $U(t)$ reaches the threshold $U_{\text{thr}}$, the neuron fires, emitting a spike, and the temporal output $H(t)$ resets to $V_{reset}$. If the membrane potential $U(t)$ does not reach the threshold, no spike is emitted, and $U(t)$ decays to $H(t)$ at a decay rate of $\beta$.

In this paper, we choose direct training with surrogate gradients as our method to train SNNs. we follow [24] to choose the arctangent-like surrogate gradients as our error estimation function when backpropagation, which regards the Heaviside step function as: $S(t) \approx \frac{1}{\pi} \arctan(\frac{\pi}{2} \alpha U(t)) + \frac{1}{2}$, where $\alpha$ is a hyper-parameter to control the frequency of the arctangent function. Therefore, the gradients of $S$ are $\frac{\partial S(t)}{\partial U(t)} = \frac{\alpha}{2} \frac{1}{(1 + (\frac{\pi}{2} \alpha U(t))^2)}$ and thus the overall model can be trained in an end-to-end manner with back-propagation through time (BPTT) [25].

## 2.2 Positional Encoding

In the field of sequential tasks, PE is crucial for models like Transformers to understand the sequential order of input tokens. Absolute PE and relative PE are two prominent methods used to incorporate positional information into these models. Absolute PE [15] assigns fixed embeddings to each position in the input sequence using trigonometric functions like sine and cosine. These embeddings are based solely on the position index and are not influenced by the token content, which are predefined and are generated as follows:

$$\text{PE}_{(pos, 2i)} = \sin\left(\frac{pos}{10000^{2i/d}}\right), \quad \text{PE}_{(pos, 2i+1)} = \cos\left(\frac{pos}{10000^{2i/d}}\right). \tag{4}$$

Here, $pos$ is the position and $d$ is the dimension. In contrast, relative PE [26–28] captures the relationships between tokens by considering their relative distances. This dynamic approach allows models to learn position-specific patterns and dependencies, which is beneficial for tasks requiring different sequence lengths or hierarchical structures.

---

[‡]Note that this is not a real relative PE. This term is adopted from the original papers.

However, existing SNN architectures reveal a notable deficiency in the integration of an effective and biologically plausible PE mechanism. As shown in Figure 1, current Transformer-based SNNs [4, 5] are primarily tailored for image classification and predominantly rely on a convolutional layer to capture the relative positional information of image patches. However, this approach resembles more of a spike-element-wise (SEW) residual connection [2] rather than a classic PE module, as it does not ensure that each image patch has a unique spike-form positional representation. Furthermore, the addition between positional spikes and the original input spikes within these models may yield hardware-unfriendly non-binary integers (i.e., neither 0 nor 1), resulting from the addition of "1" and "1". Additionally, our investigation reveals that even SNNs designed for sequential tasks, such as SpikeBERT [29, 30], SpikeGPT [31], and SpikeTCN [32], also exhibit a notable absence of an effective spike-form PE mechanism for capturing positional information.

We think that an effective PE strategy should possess the following characteristics: **uniqueness of each position** and the **capacity to capture positional information from the input data**. Furthermore, an optimal PE designed for SNNs should be **hardware-friendly** and **in spike-form**.

### 2.3 Central Pattern Generators

Central Pattern Generators (CPGs) are neural networks capable of producing rhythmic patterned outputs without sensory feedback [18, 20]. These networks are fundamental for understanding motor control in vertebrates and invertebrates and are often applied to robotics and neural control systems. Mathematically, CPGs can be modeled using systems of coupled nonlinear oscillators, and the general form can be written as:

$$\dot{\mathbf{x}} = \mathbf{F}(\mathbf{x}) + \mathbf{G}(\mathbf{x}, \mathbf{y}), \quad \dot{\mathbf{y}} = \mathbf{H}(\mathbf{y}) + \mathbf{K}(\mathbf{x}, \mathbf{y}), \tag{5}$$

where $\mathbf{x}$ and $\mathbf{y}$ are the state variables (can be seen as membrane potential) of two coupled oscillators, $\mathbf{F}$ and $\mathbf{H}$ are intrinsic dynamics of the oscillators, and $\mathbf{G}$ and $\mathbf{K}$ are the coupling functions.

## 3 Methods

In biological systems, CPGs as well as other neurons do not transmit information directly through membrane potential but through spikes. A burst of spikes will be generated only when the membrane potential of a CPG exceeds a certain threshold. Therefore, we introduced the Heaviside step function in SNN, selecting only the part that exceeds the threshold, to design the CPG-PE. In this section, we will first reveal the relationship between CPGs and PE. Then we will introduce our proposed CPG-PE and its implementations.

### 3.1 Relationship between Central Pattern Generators and Positional Encoding

Consider one of the simplest CPGs with the following assumptions:

1. The CPG is a coupled nonlinear oscillator with 2 neurons whose states are represented as $\mathbf{x}(t)$ and $\mathbf{y}(t)$.

2. Both neurons are autonomic neurons and will gain membrane voltage with constant speed, i.e., $\mathbf{F}(\mathbf{x}) = b > 0, \mathbf{H}(\mathbf{y}) = d > 0$.

3. Neuron represented by $\mathbf{x}$ will inhibits $\mathbf{y}$ while $\mathbf{y}$ excites $\mathbf{x}$. And the influence is proportional to the other neuron's state. Formally, $\mathbf{G}(\mathbf{x}, \mathbf{y}) = a\mathbf{y}, \mathbf{K}(\mathbf{x}, \mathbf{y}) = -c\mathbf{x}$ where $a > 0, c > 0$.

Now the coupled oscillators can be represented as:

$$\dot{\mathbf{x}}(t) = a\mathbf{y}(t) + b, \quad \dot{\mathbf{y}}(t) = -c\mathbf{x}(t) + d. \tag{6}$$

The general solution of this differential equation system is:

$$\mathbf{x}(t) = k_1 \cos(\sqrt{ac}\, t) + k_2 \sqrt{\frac{a}{c}} \sin(\sqrt{ac}\, t) + \frac{d}{c}, \tag{7}$$

$$\mathbf{y}(t) = -k_1 \sqrt{\frac{c}{a}} \sin(\sqrt{ac}\, t) + k_2 \cos(\sqrt{ac}\, t) - \frac{b}{a}, \tag{8}$$

where $k_1$ and $k_2$ are arbitrary constants. To simplify, we can further re-parameterize $t$ with $t' = t + \arctan(k_1/ak_2)$ as is to choose another start point, then we can rewrite Equations (7) and (8) as:

$$\mathbf{x}(t') = \sqrt{k_1^2 + \frac{a}{c}k_2^2}\sin(\sqrt{ac}\,t') + \frac{d}{c} = A_1\sin(w_1t') + b_1, \tag{9}$$

$$\mathbf{y}(t') = \sqrt{\frac{c}{a}k_1^2 + k_2^2}\cos(\sqrt{ac}\,t') - \frac{b}{a} = A_2\cos(w_2t') + b_2. \tag{10}$$

Comparing Equations (9) and (10) and Equation (4), we are astonished to find that **the PE in Transformers [15] is a particular solution of the membrane potential variations in a specific type of CPG** with properly chosen $a, b, c, d$. This finding suggests that the use of sinusoidal PE in Transformers is actually a bio-plausible choice that could possibly advance the model's ability to learn indexing and periodic information.

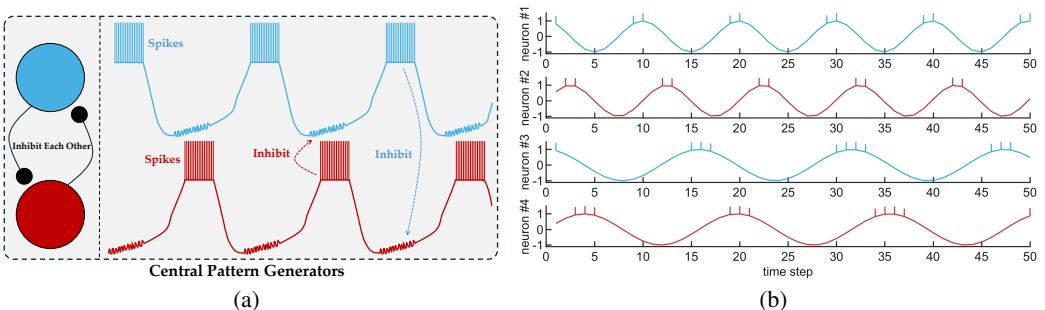

(a)                                                     (b)

Figure 2: (a) Illustration of a pair of CPG neurons demonstrating mutual inhibition through spiking activity. The spikes represent neural spikes that inhibit each other, exemplifying the coordination mechanism in CPG networks. (b) Spike trains of the first 4 CPG neurons. The curve represents the membrane potential, while the vertical lines represent spikes.

## 3.2 CPG-based Positional Encoding

Consider a system with $N$ pairs of CPG neurons, resulting in a total of $2N$ cells. Then for $i = 1, 2, ..., N$, the equations governing the CPG-PE are as follows:

$$\text{CPG-PE}^{2i-1}(t) = H\left(\cos\left(\eta\frac{t}{\tau^{\frac{i}{N}}}\right) - v^{\text{thres}}\right), \tag{11}$$

$$\text{CPG-PE}^{2i}(t) = H\left(\sin\left(\eta\frac{t}{\tau^{\frac{i}{N}}}\right) - v^{\text{thres}}\right), \tag{12}$$

where $\eta$ is a constant to control the period, $\tau$ represents the base period, and $v^{\text{thres}}$ denotes the membrane potential threshold. Note that this threshold is different from the $U_{thr}$ of spike neurons described in Equation (3). The Heaviside step function $H$ reflects a spike when the membrane potential exceeds the threshold.

It is important to clarify that the $t$ in Equation (11) and 12 is neither the time step in SNNs nor the position index. Suppose the input spike matrix $X \in \{0, 1\}^{T \times B \times L \times D}$, where $T$ is the time step in SNNs, $B$ is the batch size, $L$ is the sequence length of the input sample, $D$ is the feature dimension. To ensure the uniqueness of each position at every time step, we flatten the dimensions $T$ and $L$ into a new dimension $T \times L$. Therefore, $t$ ranges from 0 to $T \times L$. Notably, the entire CPG-PE operates in spike-form and is parameter-free. To better understand CPG-PE, we draw a simple approximation of the resulting CPG spiking patterns under the assumption of a sequence length of $L = 128$ and $N = 20$ pairs of CPG neurons, illustrated in Figure 2 (b).

## 3.3 Implementations

We design a simple implementation to apply CPG-PE to SNNs in a pluggable and hardware-friendly manner, shown in Figure 3. Before diving into the details, we want to emphasize that the data transmitted in SNNs should always be in spike-form. Therefore, the direct addition operation between two spike matrices, as used in [4, 5], should be forbidden.

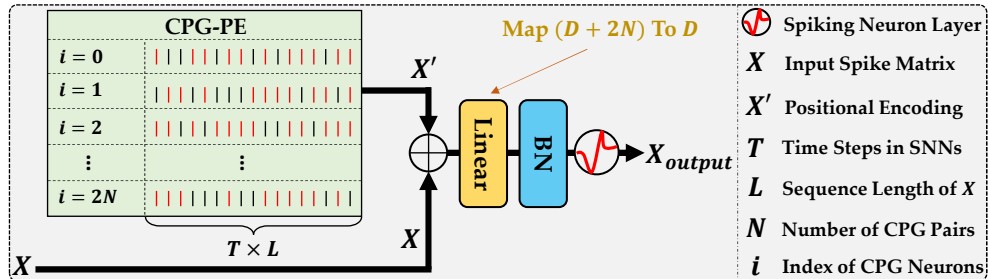

Figure 3: Illustration of applying CPG-PE to SNNs. $X$, $X'$, and $X_{output}$ are all spike matrices.

Initially, CPG-PE encodes the positional information of the input spike matrix $X$, resulting in $X'$. Then, to maintain binary values and avoid introducing non-binary elements, we opt to **concatenate** $X$ and $X'$ along the feature dimension. Lastly, a linear layer is employed to map the feature dimension from $D + 2N$ back to $D$, where $D$ is the feature dimension of $X$, and $N$ is the number of CPG pairs. This effectively neutralizes the dimensional increase caused by concatenation. The whole process can be formalized as follows:

$$X' = \text{CPG-PE}(X), \qquad X \in \{0, 1\}^{T \times B \times L \times D}, X' \in \{0, 1\}^{T \times B \times L \times 2N} \qquad (13)$$

$$X_1 = X \oplus X', \qquad X_1 \in \{0, 1\}^{T \times B \times L \times (D+2N)} \qquad (14)$$

$$X_{output} = \mathcal{SN}\left(\text{BN}\left(\text{Linear}\left(X_1\right)\right)\right), \quad X_{output} \in \{0, 1\}^{T \times B \times L \times D} \qquad (15)$$

where BN represents batch normalization and $\mathcal{SN}$ is a spike neuron layer. Furthermore, CPG-PE necessitates that input samples be sequential data, making it directly applicable to time series data and natural language. For image data, however, an adaptation is required: images must be segmented into patches similar to the approach used in the Vision Transformer [23]. Considering the compatibility with neuromorphic hardware, we also (1) implement CPG-PE with LIF neurons, and (2) integrate CPG-PE into a classic linear layer. Please refer to Appendices C and D for details.

## 4 Experiments

In this section, we conduct experiments to investigate the following research questions:
**RQ1**: Is our design of CPG-PE strategy effective and robust in sequential tasks?
**RQ2**: Can CPG-PE work well on image patches that have no inherent order?
**RQ3**: How will CPG's inner properties influence CPG-PE?
**RQ4**: Does our CPG-PE satisfy the requirements of a good PE tailored for SNNs?

### 4.1 Datasets

To assess the PE capabilities of the compared models and answer **RQ1**, we conduct two sequential tasks: **time-series forecasting**, and **text classification**. Following [32], we choose 4 real-world datasets for time-series forecasting: Metr-la [33]: This dataset contains the average traffic speed data collected from the highways in Los Angeles County. Pems-bay [33]: It consists of average traffic speed data from the Bay Area. Electricity [34]: This dataset captures hourly electricity consumption measured in kilowatt-hours (kWh). Solar [34]: It includes data on solar power production. For text classification, we follow [29] to conduct experiments on 6 benchmarks including: Movie Reviews [35], SST-2 [36], SST-5, Subj, ChnSenti, and Waimai. In addition, to answer **RQ2**, we also conduct **image classification** experiments on 1 static datasets CIFAR and 1 neuromorphic datasets CIFAR10-DVS [37]. The dataset details and metrics are provided in Appendix A.

### 4.2 Time-Series Forecasting

As discussed in Section 3.3, our proposed CPG-PE can be seamlessly integrated into any SNN capable of sequence processing. Consequently, we applied CPG-PE to the SNN counterparts of Temporal Convolutional Networks (TCN) [38], Recurrent Neural Networks (RNN) to assess the efficacy of our method in enabling SNNs to capture positional information. The results for TCN, SpikeTCN w/o PE, RNN, and Spike-RNN w/o PE are sourced from the previous study by [32]. In addition, we

Table 1: Experimental results of time-series forecasting on $4$ benchmarks with various prediction lengths $6, 24, 48, 96$. "PE" stands for positional encoding. "w/o" denotes "without" while "w/" denotes "with". The best results of **SNNs** are formatted in **bold font format**. $\uparrow$ ($\downarrow$) indicates the higher (lower) the better. Shaded ones are ours. All results are averaged across 3 random seeds.

| Model | SNN | Spike PE | Metric | Metr-la | | | | Pems-bay | | | | Solar | | | | Electricity | | | | Avg. |
|---|---|---|---|---|---|---|---|---|---|---|---|---|---|---|---|---|---|---|---|---|
| | | | | 6 | 24 | 48 | 96 | 6 | 24 | 48 | 96 | 6 | 24 | 48 | 96 | 6 | 24 | 48 | 96 | |
| TCN (ANN) | ✗ | – | $R^2\uparrow$ | .820 | .601 | .455 | .330 | .881 | .749 | .695 | .689 | .958 | .871 | .737 | .661 | .975 | .973 | .968 | .962 | .770 |
| | | | $RSE\downarrow$ | .446 | .665 | .778 | .851 | .373 | .541 | .583 | .587 | .210 | .359 | .513 | .583 | .282 | .287 | .319 | .345 | .483 |
| SpikeTCN w/o PE [32] | ✓ | – | $R^2\uparrow$ | .783 | **.603** | .468 | .326 | .811 | .729 | .662 | .633 | .937 | .840 | .708 | .650 | .970 | **.963** | .958 | .953 | .750 |
| | | | $RSE\downarrow$ | .491 | .665 | **.769** | .865 | .469 | **.541** | .625 | .635 | .259 | .401 | .541 | **.596** | .333 | **.342** | .368 | .389 | .518 |
| SpikeTCN w/ CPG-PE | ✓ | ✓ | $R^2\uparrow$ | **.802** | **.603** | .467 | **.337** | **.839** | **.737** | **.684** | **.656** | **.951** | **.861** | **.729** | **.651** | **.974** | .960 | **.959** | **.956** | **.760** |
| | | | $RSE\downarrow$ | **.469** | **.664** | .770 | **.859** | **.433** | .555 | **.604** | **.632** | **.222** | **.373** | **.521** | .606 | **.278** | .380 | **.374** | **.370** | **.506** |
| RNN (ANN) | ✗ | ✗ | $R^2\uparrow$ | .844 | .600 | .442 | .307 | .870 | .775 | .690 | .683 | .959 | .830 | .810 | .718 | .978 | .972 | .971 | .964 | .776 |
| | | | $RSE\downarrow$ | .414 | .668 | .781 | .897 | .390 | .511 | .578 | .609 | .208 | .413 | .438 | .549 | .273 | .295 | .299 | .316 | .477 |
| SpikeRNN w/o-PE [32] | ✓ | – | $R^2\uparrow$ | **.846** | **.622** | .433 | .283 | .872 | .745 | .685 | .654 | .923 | .820 | **.812** | .714 | **.977** | **.972** | .962 | **.960** | .768 |
| | | | $RSE\downarrow$ | **.412** | .648 | .794 | .935 | .387 | .528 | .588 | .634 | .278 | .425 | **.435** | .586 | .267 | .296 | .346 | .481 | .503 |
| SpikeRNN w/ CPG-PE | ✓ | ✓ | $R^2\uparrow$ | .844 | .621 | **.438** | **.306** | **.874** | **.763** | **.688** | **.667** | **.934** | **.833** | .811 | **.724** | **.977** | **.972** | **.966** | .958 | **.773** |
| | | | $RSE\downarrow$ | .416 | **.645** | **.782** | **.878** | **.380** | **.523** | **.579** | **.621** | **.264** | **.419** | **.435** | **.544** | **.265** | **.294** | **.315** | **.366** | **.482** |
| Transformer (ANN) | ✗ | ✗ | $R^2\uparrow$ | .727 | .554 | .413 | .284 | .785 | .734 | .688 | .673 | .953 | .858 | .759 | .718 | .978 | .975 | .972 | .964 | .752 |
| | | | $RSE\downarrow$ | .551 | .704 | .808 | .895 | .502 | .558 | .610 | .618 | .223 | .377 | .504 | .545 | .260 | .277 | .347 | .425 | .512 |
| Spikformer w/o PE | ✓ | – | $R^2\uparrow$ | .697 | .491 | .383 | .242 | .768 | .684 | .678 | .663 | .903 | .819 | .715 | .656 | .956 | .955 | .953 | .943 | .719 |
| | | | $RSE\downarrow$ | .581 | .753 | .828 | .917 | .521 | .607 | .613 | .627 | .319 | .439 | .548 | .602 | .371 | .375 | .386 | .450 | .559 |
| Spikformer w/ RPE [4] | ✓ | ✓ | $R^2\uparrow$ | .713 | .527 | .399 | .267 | .773 | .697 | .686 | .667 | .929 | .828 | .744 | .674 | .959 | .955 | .955 | .954 | .733 |
| | | | $RSE\downarrow$ | .565 | .725 | .818 | .903 | .514 | .594 | .606 | .621 | .272 | .426 | .519 | .586 | .373 | .371 | .379 | .382 | .541 |
| Spikformer w/ Float-PE | ✓ | ✗ | $R^2\uparrow$ | .699 | .502 | .409 | .255 | .762 | .704 | .687 | .666 | .934 | .834 | .752 | .699 | .970 | .967 | .960 | .957 | .734 |
| | | | $RSE\downarrow$ | .578 | .744 | .810 | .912 | .527 | .588 | .605 | .623 | .264 | .418 | .512 | .563 | .307 | .322 | .356 | .362 | .531 |
| Spikformer w/ CPG-PE | ✓ | ✓ | $R^2\uparrow$ | **.726** | .526 | **.419** | **.287** | **.780** | .712 | **.690** | .666 | **.937** | .833 | **.757** | .707 | **.972** | **.970** | .966 | **.960** | **.744** |
| | | | $RSE\downarrow$ | **.553** | .720 | **.806** | **.890** | .508 | .580 | **.602** | .622 | **.257** | .420 | **.506** | .555 | **.299** | .310 | .314 | **.355** | **.519** |
| Spikformer w/ CPG-Full | ✓ | ✓ | $R^2\uparrow$ | .719 | **.530** | .417 | .286 | .779 | **.714** | .689 | **.668** | .936 | **.835** | **.757** | **.709** | .971 | **.971** | **.968** | .962 | **.744** |
| | | | $RSE\downarrow$ | .560 | **.719** | .807 | .893 | **.507** | **.577** | .605 | **.620** | .260 | **.417** | .508 | **.548** | .304 | **.308** | **.311** | .439 | .523 |

deliberately conducted experiments on PE in Spikformer to explore whether our specially designed CPG-PE is truly more suitable for SNNs than all previous PEs. Notably, we also investigated the modularization of CPG, i.e., replacing all Linear layers with CPG-Linear layers (See Appendix D), and its impact on the Spikformer model for time-series forecasting, i.e., Spikformer w/ CPG-Full. We report the results on $4$ time-series forecasting benchmarks with various prediction lengths in Table 1. We also list results from ANNs for reference.

In summary, the results presented in Table 1 indicate that SNNs equipped with the CPG-PE module significantly outperform their counterparts lacking the PE feature. This finding effectively addresses **RQ1** from a time-series analysis perspective. Detailed findings include:

**(1) CPG-PE enables SNNs to successfully capture positional information**. SNNs, including models such as Spike-TCN, Spike-RNN, and Spikformer, when integrated with CPG-PE, show superior performance compared to those without PE. Notably, CPG-PE also reduces the performance disparity between SNNs and traditional ANNs in time-series forecasting tasks, evidenced by an average increase of $0.013$ in $R^2$ and a decrease of $0.022$ in RSE.

**(2) CPG-PE is the most suitable position encoding strategy for Spikformer**. In addition to CPG-PE, other encoding strategies such as Float-PE (the original PE in Transformer) and RPE (the original PE in Spikformer) were also evaluated. The Spikformer equipped with CPG-PE emerged as the top-performing variant, confirming CPG-PE as the most suitable PE strategy for SNNs.

**(3) CPG-Full module can also effectively model the positional information of time series data**. The CPG-Full module's performance in modeling positional information of time-series data is comparable to that of CPG-PE, with average $R^2$ values nearly identical to those of Spikformer with CPG-PE and significantly better than those of other models.

### 4.3 Text Classification

In addition to time-series forecasting, natural language processing (NLP) serves as another critical domain to assess the efficacy of the CPG-PE module in encoding positional information. Following the pioneering work of [29], who first employed Spikformer for text classification tasks, we extended this application to 6 benchmark datasets. We also include results from fine-tuned BERT for reference.

Table 2: Accuracy on 6 text classification benchmarks. The best results of SNNs and ANNs are formatted in bold font format. Experimental results are averaged across 5 random seeds.

| Model | SNN | Spike PE | Param (M) | English Dataset | | | | Chinese Dataset | | Avg. |
|---|---|---|---|---|---|---|---|---|---|---|
| | | | | MR | SST-2 | Subj | SST-5 | ChnSenti | Waimai | |
| Fine-tuned BERT [39] | ✗ | ✗ | 109.8 | **87.63**±0.18 | **92.31**±0.17 | **95.90**±0.16 | **50.41**±0.13 | **89.48**±0.16 | **90.27**±0.13 | **84.33** |
| Spikformer w/o PE [29] | ✓ | – | 109.8 | 75.87±0.35 | 81.71±0.31 | 91.60±0.30 | 41.84±0.39 | 85.62±0.25 | 86.87±0.28 | 77.25 |
| Spikformer w/ Random-PE | ✓ | ✓ | 110.4 | 75.90±0.42 | 81.64±0.31 | 91.40±0.35 | 41.86±0.41 | 85.63±0.29 | 86.90±0.30 | 77.23 |
| Spikformer w/ Float-PE | ✓ | ✗ | 109.8 | 79.67±0.36 | 82.18±0.34 | 92.20±0.31 | 42.58±0.41 | 85.71±0.26 | 88.34±0.32 | 78.44 |
| Spikformer w/ CPG-PE [Ours] | ✓ | ✓ | 110.4 | **82.42**±0.42 | **82.90**±0.33 | **92.50**±0.25 | **43.62**±0.36 | **86.54**±0.26 | **88.49**±0.29 | **79.41** |

The results presented in Table 2 shows that Spikformer enhanced with CPG-PE achieves the state-of-the-art performance across 6 benchmarks, effectively addressing **RQ1**. Meanwhile, we conducted a set of ablation experiments to eliminate the effects of increased parameter counts on model performance. Specifically, we replaced the spike-form positional encoding matrix obtained from CPG with a randomly generated spike matrix (See "Spikformer w/ Random PE" Row). By comparing these two configurations, we confirmed the effectiveness of our proposed CPG-PE.

## 4.4 Image Classification

In this section, we aim to answer **RQ2**. To adapt the CPG-PE for image classification, it is essential to conceptualize the array of image patches as sequential data. Consequently, some SNN models that do not incorporate a concept of "sequence length" in their spike matrices, such as SEW-Resnet [2], are incompatible with the integration of a CPG-PE module. Therefore, we only consider ViT-liked SNN, i.e. Spikformer, in this experiment. We also include results from ViTs for reference.

Table 3: Evaluation on image classification benchmarks. Float-PE denotes the original PE of the Transformer, while RPE denotes the original PE of the Spikformer. Numbers with $^*$ denote our implementation. The best results of SNNs and ANNs are formatted in bold font format. All results are averaged across $4$ random seeds.

| Model | SNN | Spike PE | CIFAR10 | | CIFAR10-DVS | | CIFAR100 | | Avg. |
|---|---|---|---|---|---|---|---|---|---|
| | | | Param (M) | Accuracy | Param (M) | Accuracy | Param (M) | Accuracy | |
| Vision-Transformer [23] | ✗ | ✗ | 9.32 | **96.73** | – | – | 9.36 | **81.02** | – |
| Spikformer w/o PE | ✓ | – | 8.00 | 93.77 | 1.99 | 76.40 | 8.04 | 73.59 | 81.25 |
| Spikformer w/ Random-PE | ✓ | ✓ | 8.17 | 93.85 | 2.06 | 76.44 | 8.20 | 73.54 | 81.27 |
| Spikformer w/ Float-PE | ✓ | ✗ | 8.00 | 94.42 | 1.99 | 77.60 | 8.04 | 74.73 | 82.25 |
| Spikformer w/ RPE [4] | ✓ | ✓ | 9.33 | 94.64* | 2.57 | 77.95* | 9.37 | 76.78* | 83.12 |
| Spikformer w/ CPG-PE [Ours] | ✓ | ✓ | 8.17 | **94.82** | 2.06 | **78.06** | 8.20 | **77.27** | **83.38** |
| Spikformer w/ CPG-PE [Equal Param] | ✓ | ✓ | 7.99 | 94.60 | 1.99 | 78.00 | 8.02 | 76.91 | 83.17 |

We report the parameter counts and classification accuracy in Table 5. To elaborate, Spikformer with CPG-PE outperforms other variants, demonstrating the effectiveness of CPG-PE even when the sequence is an array of image patches lacking inherent order. Notably, owing to our streamlined implementation, the parameter count for Spikformer with CPG-PE is significantly reduced compared to the original Spikformer w/ RPE [4], with a reduction of $1.16$ M. What's more, we conducted ablation experiments on model parameters by reducing the parameter count of Spikformer with CPG-PE to be comparable to Spikformer w/o PE, allowing for a more direct performance comparison, as shown in the last line in Table 5. The results on ImageNet are reported in Appendix E.

However, it is essential to acknowledge that the improvements in image classification are relatively modest compared to those observed in time series and text applications. This phenomenon can largely be attributed to the intrinsic *non-ordered nature* of image patches. Unlike text or time series data, where sequential order is crucial and inherently informative, image patches do not possess a natural or fixed sequence. This lack of order means that traditional methods of positional encoding, which significantly benefit ordered data by providing contextual positioning, are less effective. Thus, the application of our positional encoding techniques, optimized for data with inherent sequential order, does not translate as effectively to the domain of image classification.

## 4.5 Sweeping CPG properties

In this section, we investigate the influence of CPG properties on the ability to model positional information, addressing **RQ3**. To this end, we evaluated the Spikformer model with CPG-PE by varying the base period $\tau$ and the number of CPG pairs $N$ (see Equations (11) and (12)) in time-series forecasting and image classification tasks.

From Figure 4 (a) and (b), we observe that CPG-PE is insensitive to the base period $\tau$ (in biological neurons, $\tau$ is affected by the physiological properties of the CPG circuit such as RC constant and synaptic delay). The sequence lengths ($T \times L$) of the time series and image patches are no larger than 672 ($4 \times 168$) for all benchmarks, preventing repetitions in CPGs. Therefore, when $N = 20$, sweeping $\tau \in \{100, 1000, 5000, 10000\}$ makes minor influence on performance. Furthermore, Figure 4 (c) and (d) demonstrate that when $\tau = 10000$, increasing the number of CPG pairs $N$ enhances Spikformer's performance. This is reasonable because more CPG neurons reduce repetitions in positional representations of $X'$.

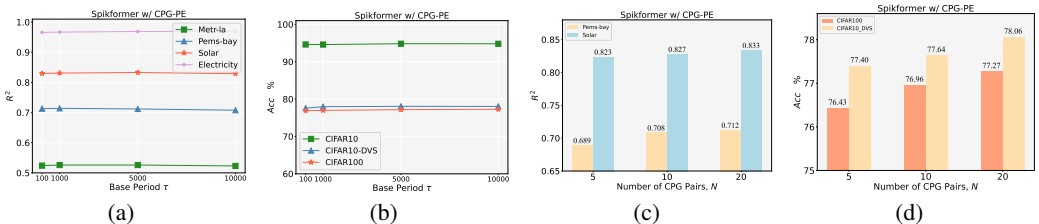

Figure 4: (a)(c) $R^2$ versus $\tau$ and $N$ on time-series forecasting tasks. (b)(d) Accuracy versus $\tau$ and $N$ on image classification tasks. $\tau \in \{100, 1000, 5000, 10000\}$, $N \in \{5, 10, 20\}$.

## 4.6 Positional Encoding Analysis

In this section, we want to address **RQ4**. As mentioned in Section 2.2, an ideal PE method for SNNs should include the following characteristics: (1) **Uniqueness of each position**; (2) **Compatibility with neuromorphic hardware**; (3) **Formulation in spike-form**. Our implementations ensure compatibility with neuromorphic hardware (2), and the CPG-PE is inherently formulated in spike-form, satisfying (3). Therefore, in order to assess (1), the uniqueness of each position, we would like to compare the CNN-based RPE in [4, 5] and our proposed CPG-PE, focusing specifically on their capacity to provide distinct positional signals. This analysis was conducted using the CIFAR10-DVS dataset, where we calculated the repetition rate of spike positional representations across all positions. Our findings were notable: the positional spike matrices produced by RPE exhibited a repetition rate as high as **12.19**%, which significantly undermines its effectiveness for PE. In contrast, our proposed CPG-PE exhibited no repetition, demonstrating that our CPG-PE is well-suited for serving as the PE module in SNNs. Please refer to Appendix B for details.

## 5 Related Work

### 5.1 Spike Encoding Methods

Spiking neural networks employ several coding methods to encode input information, each offering unique advantages. Direct coding [5, 40], the simplest form and widely-used in image tasks, directly associates spikes with specific values or events, providing straightforward and interpretable outputs but often lacking efficiency for complex tasks. Rate coding [8, 41], where the input is represented by the frequency of spikes within a given timeframe, is more robust and widely used but can be less precise due to its reliance on averaged spike rates. Temporal coding (a.k.a latency coding) [42, 43] encodes information based on the timing of individual spikes, allowing for high temporal precision and efficient representation of dynamic inputs, though it can be computationally demanding. In addition, delta coding [44] represents changes in input signals through spikes, focusing on differences rather than absolute values, which can enhance efficiency and response times but may introduce complexity in decoding. Each of these methods contributes to the versatility and applicability of SNNs in various domains, from neuroscience to artificial intelligence. The SNNs we considered in this paper should fall into the category of rate coding since back-prop is conducted on spike rate. Meanwhile, CPG-PE can be considered converting temporal information into spike rate of a group of neurons (Equations 11 and 12), and this is why CPG-PE can improve performance for sequential data. It is possible to introducing learning algorithms of temporal coding for the CPG neurons to tackle more complex sequence structure, which remains as future work.

### 5.2 Positional Encoding in SNNs

Currently, few works have demonstrated the importance of PE approaches in SNNs. Spikformer [4] and Spike-driven Transformer [5] utilize a combination of "one convolutional layer + one batch normalization layer + one spiking neuron layer " to generate learnable "relative positional encoding". From our perspective, this strategy is more like a spike-element-wise residual connection [2], rather than a classic PE module. The unique representation of each position is a fundamental requirement for a robust PE module. However, the spike position matrix generated by their method may result in the same spike representation for different positions. Additionally, the addition of the input spike matrix and the position spike matrix will result in the occurrence of non-binary numbers (i.e., 2) due

to the addition of 1 and 1. For spiking graph neural networks, [45] proposed learnable positional graph spikes, aiming to capture neighbor information within graphs rather than sequences. Therefore, drawing inspiration from the periodic automatic spike generation pattern of CPGs, we propose a biologically plausible and effective spike-form absolute position encoding method called CPG-PE.

## 6 Rethinking the Role of CPGs in Neuroscience

Our study also provides novel insights into neuroscience on understanding the role of CPG in nervous systems. While traditionally CPG is believed to play a crucial role in producing the rhythmic motor patterns necessary for locomotion and other repetitive movements [18, 20], the analogy to PE in this work reveals that CPG can make a significant contribution in processing sequential data by encoding the positional information into unique spiking patterns at different times. This does not only work for time-series sensory input like auditory signals but also for visual sensory data: e.g. when a person looks at an image, saccades (eye movements) allow retinal neurons to receive different parts of the image at different times. This indicates that CPG neurons could potentially be utilized to encode positional information. Another extensive thought is that as PE can be learnable in ANNs, CPG may also benefit from adaptability to the data [46]. The hypothesis, however, remains to be examined through neuroscientific experiments [47].

## 7 Conclusion

In conclusion, inspired by central pattern generators, we introduce a pioneering position encoding approach termed CPG-PE, specifically tailored to mitigate the constraints associated with current PE techniques within SNNs. We mathematically prove that abstract PE in the Transformer is a particular solution of the membrane potential variations in a specific type of CPG. Furthermore, through comprehensive empirical investigations across diverse domains including time-series forecasting, natural language processing, and image classification, we demonstrate that the CPG-PE satisfies all the requirements of PE tailored for SNNs. The limitations and future work are discussed in Appendix F.

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

## Acknowledgments

The authors would like to thank the anonymous reviewers for their valuable comments. This work was supported partially by National Natural Science Foundation of China (No. 62076068).

## Broader Impact

This work aims to advance the field of spiking neural networks (SNNs). Unlike artificial neural networks (ANNs) which have been applied widely in people's lives, SNNs are still undergoing fundamental research. We do not see any negative societal impacts of this work.

## Reproducibility Statement

The authors have diligently worked to ensure the reproducibility of the empirical results presented in this paper. The datasets, experimental setups, evaluation metrics, and hyperparameters are thoroughly described in Appendices A and B. Furthermore, the source code for the proposed PE method has been available at `https://github.com/microsoft/SeqSNN`.

## A    Datasets

### A.1    Time-series Forecasting

Detailed statistical characteristics and distribution ratios for each dataset are provided in the following:

Table 4: The statistics of time-series datasets.

| Dataset | Samples | Variables | Observation Length | Train-Valid-Test Ratio |
|---|---|---|---|---|
| Metr-la | $34,272$ | 207 | 12, (short-term) | $(0.7, 0.2, 0.1)$ |
| Pems-bay | $52,116$ | 325 | 12, (short-term) | $(0.7, 0.2, 0.1)$ |
| Solar-energy | $52,560$ | 137 | 168, (long-term) | $(0.6, 0.2, 0.2)$ |
| Electricity | $26,304$ | 321 | 168, (long-term) | $(0.6, 0.2, 0.2)$ |

### A.2    Text Classification

Here are the datasets we used in text classification experiments:

- **MR**: MR, which stands for Movie Review, is a dataset containing movie-review documents labeled based on their overall sentiment polarity (positive or negative) or subjective rating [35].
- **SST-5**: SST-5 includes $11,855$ sentences from movie reviews for sentiment classification across 5 categories: very negative, negative, neutral, positive, and very positive [36].
- **SST-2**: SST-2 is the binary version of SST-5, containing only 2 classes: positive and negative.
- **Subj**: The Subj dataset is designed to classify sentences as either subjective or objective[*].
- **ChnSenti**: ChnSenti consists of approximately $7,000$ Chinese hotel reviews, each annotated with a positive or negative label[†].
- **Waimai**: This dataset contains around $12,000$ Chinese user reviews from a food delivery platform, intended for binary sentiment classification (positive and negative)[‡].

### A.3    Image Classification

Here are the datasets we used in image classification experiments: CIFAR dataset comprises a collection of $60,000$ images, partitioned into $50,000$ training and $10,000$ testing images, each with a

---

[*]`https://www.cs.cornell.edu/people/pabo/movie-review-data/`

[†]`https://raw.githubusercontent.com/SophonPlus/ChineseNlpCorpus/master/datasets/ChnSentiCorp_htl_all/`
`ChnSentiCorp_htl_all.csv`

[‡]`https://raw.githubusercontent.com/SophonPlus/ChineseNlpCorpus/master/datasets/waimai_10k/waimai_10k.csv`

resolution of $32 \times 32$ pixels. The CIFAR10-DVS dataset represents a neuromorphic adaptation of this original set, where static images have been transformed to accommodate the recording capabilities of a Dynamic Vision Sensor (DVS) camera. This conversion results in a dataset consisting of $9,000$ training samples and $1,000$ test samples with $128 \times 128$ resolution.

## B  Experiment Settings

### B.1  Time-series Forecasting

**Metrices**    The metrics we used in time-series forecasting are the coefficient of determination ($R^2$) and the Root Relative Squared Error (RSE).

$$R^2 = \frac{1}{MCL} \sum_{m=1}^{M} \sum_{c=1}^{C} \sum_{l=1}^{L} \left[ 1 - \frac{(Y_{c,l}^m - \hat{Y}_{c,l}^m)^2}{(Y_{c,l}^m - \bar{Y}_{c,l})^2} \right], \tag{16}$$

$$\text{RSE} = \sqrt{\frac{\sum_{m=1}^{M} ||\mathbf{Y}^m - \hat{\mathbf{Y}}^m||^2}{\sum_{m=1}^{M} ||\mathbf{Y}^m - \bar{\mathbf{Y}}||^2}}. \tag{17}$$

In these formulas, $M$ symbolizes the size of the test sets, $C$ denotes the number of channels, and $L$ signifies the length of predictions. $\bar{\mathbf{Y}}$ represents the average of $\mathbf{Y}^m$. The term $Y_{c,l}^m$ refers to the $l$-th future value of the $c$-th variable for the $m$-th sample, while $\bar{Y}_{c,l}$ indicates the mean of $Y_{c,l}^m$ across all samples. The symbols $\hat{\mathbf{Y}}^m$ and $\hat{Y}_{c,l}^m$ are used to represent the ground truth values. Compared to Mean Squared Error (MSE) or Mean Absolute Error (MAE), these metrics offer greater resilience against the absolute values of the datasets, making them particularly useful in the time-series forecasting setting.

**Model Architecture**    All SNNs take 4 time steps. For SpikeTCNs and SpikeRNNs, we follow the same settings as [32]. We construct all Spikformer as 2 blocks, setting the feature dimension as 256, and the hidden feature dimension in FFN as 1024. For CPG-PE settings, we set $\tau = 10000.0$, $N = 20$, $\eta = 1$, and $v^{\text{thres}} = 0.8$.

**Training Hyper-parameters**    we set the training batch size as 64 and adopt Adam [48] optimizer with a cosine scheduler of learning rate $1 \times 10^{-4}$. An early stopping strategy with a tolerance of 30 epochs is adopted. We conducted time-series forecasting experiments on 24G-V100 GPUs. On average, a single experiment takes about 1 hour under the settings above.

### B.2  Text Classification

**Model Achirecture**    All models are with 12 encoder blocks and 768 feature embedding dimension. It is important to note that the original implementation of [29] incorporates a layer normalization module that poses challenges to hardware compatibility. To address this, we have substituted layer normalization with batch normalization in our directly-trained Spikformer models for text classification tasks. For CPG-PE settings, we set $\tau = 10000.0$, $N = 20$, $\eta = 1$, and $v^{\text{thres}} = 0.8$.

**Training Hyper-parameters**    We directly trained Spikformers with arctangent surrogate gradients on all datasets. We use the BERT-Tokenizer in Huggingface§ to tokenize the sentences to token sequences. We pad all samples to the same sequence length of 256. We conducted text classification experiments on 4 RTX-3090 GPUs, and set the batch size as 32, optimizer as AdamW [49] with weight decay of $5 \times 10^{-3}$, and set a cosine scheduler of starting learning rate of $5 \times 10^{-4}$. What's more, in order to speed up the training stage, we adopt the automatic mixed precision training strategy. On average, a single experiment takes about 1.5 hours under the settings above.

### B.3  Image Classification

**Model Architecture**    For all Spikformer models, we standardized the configuration to include 4 time steps. Specifically, for the CIFAR10 and CIFAR100 datasets, the models were uniformized

---

§`https://huggingface.co/`

with $4$ encoder blocks and a feature embedding dimension of $384$. For the CIFAR10-DVS dataset, the models were adjusted to have $2$ encoder blocks and a feature embedding dimension of $256$. For CPG-PE settings, we set $\tau = 10000.0$, $N = 20$, $\eta = 2\pi$, and $v^{\text{thres}} = 0.8$.

**Training Hyper-parameters** We honestly follow the experimental settings in [4], whose source code and configuration files are available at `https://github.com/ZK-Zhou/spikformer`. As the training epochs are quite big ($300$ epochs) in their settings, we choose to use one 80G-A100 GPU, and it takes about $3$ hours to conduct a single experiment, on average.

### B.4 Details about Positional Encoding Analysis

We conducted positional encoding analysis experiments on the CIFAR10-DVS dataset. For the original Spikformer with relative positional encoding (RPE) as described by [4], the input and output channels of Conv2d are both set to $384$. In our Spikformer with CPG-PE, the parameters are set to $\tau = 10000.0$ and $N = 20$. Given the time step $T = 4$ and the sequence length $L = 160$ for the image patches in CIFAR10-DVS samples, the total "length" $T \times L$ in CPG-PE is $640$. We then calculated the repetition rate of positions. The results showed that the repetition rate for RPE is $12.19\%$, whereas for CPG-PE, it is $0.00\%$.

## C  Implement CPG-PE with LIF Neurons

In this section, we demonstrate that CPG-PE is a hardware-friendly design. While implementing the sinusoidal potential on the neuromorphic chips is not challenging (e.g., by maintaining additional LC circuits), we show how a CPG-PE neuron can be physically implemented with only 2 LIF neurons defined by Equations (1) to (3) and thus introducing no extra efforts on chip designs.

A CPG-PE neuron, after discretization, can be viewed as an autonomic neuron that will emit a burst of $K$ spikes after resting for $R$ time steps. The key idea is to set two LIF neurons, namely the *Emitter* and the *Resetter*. The emitter will draw constant current from the source, and as soon as its membrane potential reaches the threshold after $R$ time steps, it will start emitting spikes constantly until receiving the reset signal from the resetter. The resetter, which will remain at the resting potential until it receives signals from the emitter, will count the number of spikes and emit a reset signal (inhibition signal) to the emitter after receiving $K$ spikes.

We first prove the following Lemma, which establishes the relationship between the start time of the first spike and a constant input current.

**Lemma 1.** *Given an LIF neuron defined by Equations* (1) *to* (3) *with decay rate $\beta$ and threshold $U_{thr}$, starting with resting potential $U(0) = 0$, if fed with the constant current $I(t) = I_c > 0$, the first spike will emit at:*

$$T_{min} = \left\lceil \log_\beta(\beta - \frac{U_{thr}\beta(1-\beta)}{I_c}) \right\rceil. \tag{18}$$

*Proof.* By definition, before the time to emit the first spike, we have $S(t) = 0$. Thus Equation (1) can be rewrite as:

$$U(k\Delta t) = \beta U((k-1)\Delta t) + I_c. \tag{19}$$

Simplifying the recurrence relation, we can obtain:

$$U(k\Delta t) = \frac{I_c}{\beta}\left(\frac{1-\beta^k}{1-\beta} - 1\right). \tag{20}$$

The first spike is generated when $U(k\Delta t) \leq U_{thr}$, thus we have:

$$k \geq \log_\beta(\beta - \frac{U_{thr}\beta(1-\beta)}{I_c}), \tag{21}$$

that is to say,

$$T_{min} = \left\lceil \log_\beta(\beta - \frac{U_{thr}\beta(1-\beta)}{I_c}) \right\rceil. \tag{22}$$

$\square$

Now we can implement the CPG-PE with LIF neurons:

**Theorem 1.** *Given 2 LIF neurons, the emitter and the resetter, with decay rate $\beta$, threshold $U_{thr}$, and reset potential $V_{reset}$, starting with resting potential $U_e(0) = U_r(0) = 0$. If*

$$I_e(t) = I_{c1} + S_e(t - \Delta t)(U_{thr} - I_{c1} - V_{reset}) - S_r(t - \Delta t)U_{thr}, \tag{23}$$

$$I_r(t) = S_e(t - \Delta t)I_{c2} - S_r(t - \Delta t)(I_{c2} + V_{reset}), \tag{24}$$

$$I_{c1} = \frac{U_{thr}\beta(1 - \beta)}{\beta - \beta^R}, \tag{25}$$

$$I_{c2} = \frac{U_{thr}\beta(1 - \beta)}{\beta - \beta^{K-1}}, \tag{26}$$

*then the system will have the period of $T = (R + K)\Delta t$, and $\forall i \in \mathbb{N} \cap [0, R + K - 1], k \in \mathbb{N}$:*

$$S_e(i\Delta t + kT) = \begin{cases} 0, & 0 \le i < R, \\ 1, & R \le i < R + K. \end{cases} \tag{27}$$

*Proof.* Assuming the first spike generated by the emitter emits at time step $T_1$. For every $0 \le t < T_1$, we have:

$$S_e(t) = S_r(t) = 0, \tag{28}$$

$$I_e(t) = I_{c1}, I_r(t) = 0. \tag{29}$$

Since the input current of the emitter is a constant, by Lemma 1, we immediately get:

$$T_1 = \left\lceil \log_\beta(\beta - \frac{U_{thr}\beta(1 - \beta)}{I_{c1}}) \right\rceil = R. \tag{30}$$

Starting from time step $R$, let's assume the first spike generated by the resetter emits at time step $T_2$. Then for every $T_1 \le t < T_2$, we have:

$$S_e(t) = 1, S_r(t) = 0, \tag{31}$$

$$I_e(t) = U_{thr} - V_{reset}, I_r(t) = I_{c2}. \tag{32}$$

Starting from $T_1$, for the emitter, the input current allows a spike event for every time step. And the input current of the resetter is a constant. Again, by applying Lemma 1, we can get:

$$T_2 = T_1 + \left\lceil \log_\beta(\beta - \frac{U_{thr}\beta(1 - \beta)}{I_{c2}}) \right\rceil = R + K - 1. \tag{33}$$

Now Consider the state at time step $R + K$:

$$S_e((R + K - 1)\Delta t) = S_r((R + K - 1)\Delta t) = 1, \tag{34}$$

$$I_e((R + K)\Delta t) = I_r((R + K)\Delta t) = -V_{reset}, \tag{35}$$

$$U_e((R + K)\Delta t) = U_r((R + K)\Delta t) = 0. \tag{36}$$

This is the same as the membrane potential at time step 0. Therefore, the system will behave periodically with period $T = (R + K)\Delta T$. $\square$

Theorem 1 gives a possible CPG-PE design with 2 LIF neurons, with the emitter generating $K$ consecutive spikes every $R + K$ time steps. This demonstrates that incorporating CPG-PE into the current SNN architecture is completely bio-plausible and will not introduce any burden of redesigning hardware.

## D   Implement CPG-Linear

We have developed a simple modularization implementation to integrate our proposed CPG-PE with original linear layers, as depicted in Figure 3 (b). Consider the original linear layer's input and output dimensions as $D_{in}$ and $D_{out}$, respectively. Our objective is to incorporate CPG-PE within this framework. Following the application of the CPG-PE module, the modified input $X'$ is obtained. $X$ is then input into $\text{Linear}_1$ and $X'$ into $\text{Linear}_2$, resulting in outputs $X_1$ and $X_2$, respectively. Both

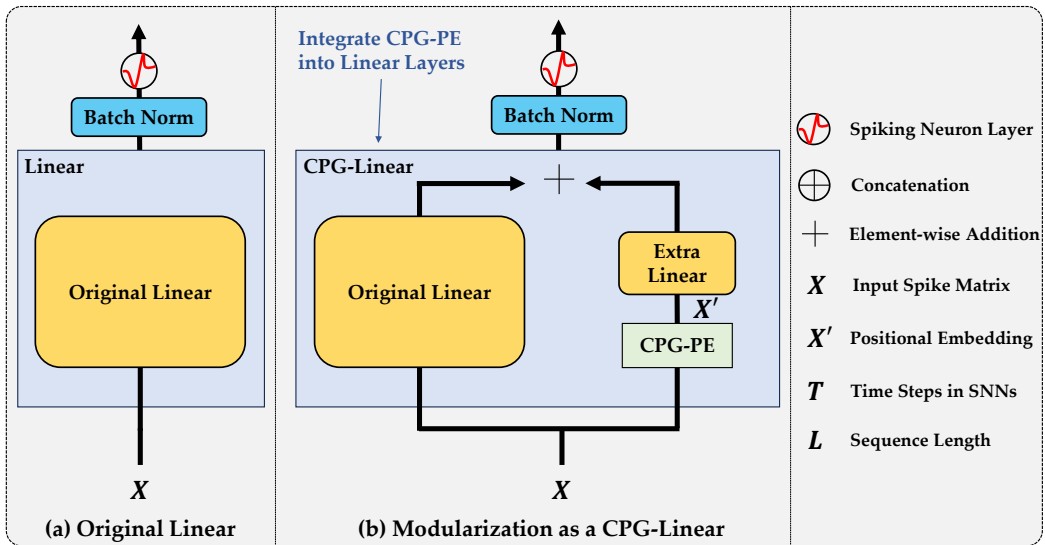

Figure 5: An illustration of the implementation of integrating a CPG-PE into a linear layer.

$\text{Linear}_1$ and $\text{Linear}_2$ maintain an output dimension of $D_{out}$. The final step involves summing $X_1$ and $X_2$ to produce $X_3$, which is subsequently processed through batch normalization (BN) and spike normalization ($\mathcal{SN}$). We term this implementation as "CPG-Linear" and formulize as follows:

$$X' = \text{CPG-PE}(X), \qquad X \in \{0,1\}^{T \times B \times L \times D_{in}}, X' \in \{0,1\}^{T \times B \times L \times 2N} \quad (37)$$

$$X_1 = \text{Linear}_1(X), X_2 = \text{Linear}_2(X'), \quad X_1, X_2 \in \mathbb{R}^{T \times B \times L \times D_{out}} \quad (38)$$

$$X_3 = X_1 + X_2, \qquad X_3 \in \mathbb{R}^{T \times B \times L \times D_{out}} \quad (39)$$

$$X_{output} = \mathcal{SN}\left(\text{BN}\left(X_3\right)\right), \qquad X_{out} \in \{0,1\}^{T \times B \times L \times D_{out}} \quad (40)$$

where $+$ denotes element-wise addition. This implementation described above is fundamentally identical to Figure 3, within the context of a single linear layer. However, the CPG-Linear can seamlessly replace **any linear layer** in SNNs.

## E Results on ImageNet

We have conducted experiments with Spikformer without positional encoding (PE), Spikformer with relative positional encoding (RPE), and Spikformer with our proposed CPG-PE on the ImageNet dataset. The results are as follows:

Table 5: Evaluation on ImageNet benchmarks. We employed 8 encoder blocks and 384 feature embedding dimensions across all models.

| Model | SNN | Spike PE | ImageNet | |
|---|---|---|---|---|
| | | | Param (M) | Accuracy |
| Spikformer w/o PE | ✓ | – | 15.50 | 69.46 |
| Spikformer w/ RPE [4] | ✓ | ✓ | 16.81 | 70.24 |
| Spikformer w/ CPG-PE [Ours] | ✓ | ✓ | 15.66 | **71.17** |

Specifically, we set the depth to 8 and the dimension of representation to 384. From the table, we can see that CPG-PE performs well on large-scale image datasets. We believe that the above results demonstrate the effectiveness of our proposed CPG-PE in positional encoding.

## F Limitations and Future Works

In this section, we will discuss the limitations and future works of our paper.

### F.1 Limitations

As mentioned in Section 3.3, our CPG-PE can not be directly applied to those SNNs where spike matrices do not have a "sequence length" dimension. Our CPG-PE is optimized for processing sequential data, making it ideal for applications involving time series or natural language. This intrinsic design, however, does not naturally extend to image data, which typically benefits from direct convolutional operations that capture spatial relationships across the entire image dimensions—height and width. In contrast, CPG-PE requires the segmentation of images into patches, a method inspired by the Vision Transformer. This adaptation contrasts with approaches like the Convolutional 2D layer, which applies convolution operations directly across the height and width of an image without requiring segmentation into smaller, discrete patches. The necessity to adapt CPG-PE for image data through patching can introduce complexities and potential performance bottlenecks, as it may not effectively capture the continuous spatial relationships and local features in the image, which are crucial for tasks such as object recognition and scene understanding.

### F.2 Future Works

To enhance the applicability of the CPG-PE model to a broader range of data types, especially image data, future research could focus on developing a hybrid model that integrates the strengths of CPG-PE with traditional convolutional layers. This integration could potentially allow the model to handle both sequential and spatial data efficiently without the need for extensive pre-processing or adaptation. Specifically, integrating direct convolution operations that work across the entire spatial dimensions of an image within the CPG-PE architecture could help preserve spatial relationships and improve feature extraction capabilities. Additionally, exploring the use of adaptive patch sizes or dynamically adjusting the patching mechanism based on the nature of the input data could also provide a more flexible and performance-optimized approach. These advancements would make the model more versatile and capable of tackling a wider array of tasks across different domains.

Additionally, considering that CPG-PE is an absolute positional encoding designed for SNNs, it could be beneficial to explore the potential of implementing learnable relative positional encodings in SNNs. Such encodings would need to be developed to meet specific criteria: they must maintain the spike-form characteristic essential to SNNs and ensure the uniqueness of each position's encoding. This approach could significantly enhance the model's ability to capture and utilize the temporal dynamics of input data more effectively, potentially leading to more nuanced and context-aware processing capabilities. Exploring adaptive patch sizes or dynamically adjusting the patching mechanism based on the nature of the input data could also provide a more flexible and performance-optimized approach. These advancements would make the model more versatile and capable of tackling a wider array of tasks across different domains.

