# OpenReview forum: "Advancing Spiking Neural Networks for Sequential Modeling with Central Pattern Generators"
_NeurIPS.cc/2024/Conference — NeurIPS 2024 spotlight_

### Official Review · Reviewer_Pjez · 2024-06-14

**Soundness:** 4
**Presentation:** 3
**Contribution:** 4
**Rating:** 7
**Confidence:** 5

**Summary:**

This paper proposes a new position encoding technique CPG-PE for SNNs, which is inspired by the central pattern generator in the human brain and improves the ability of SNNs to process sequence data. Experimental results show that CPG-PE outperforms traditional SNNs in multiple fields such as time series prediction, text classification and image classification.

**Strengths:**

1. The bio-inspired CPG-PE technique is proposed to enhance the sequence processing capability of SNNs.
2. The effectiveness of CPG-PE is verified in multiple fields, including time series, text classification, and image classification.
3. The performance of SNNs in various tasks is significantly improved, outperforming traditional models.
4. CPG-PE is designed with compatibility with neuromorphic hardware in mind, facilitating deployment in practical applications.

**Weaknesses:**

1. No experiments were conducted on the large-scale image classification Imagenet dataset.
2. Other encoding methods in SNN such as rate coding [1] and temporal coding [2;3], were not discussed.

[1] Kim Y, Park H, Moitra A, et al. Rate coding or direct coding: Which one is better for accurate, robust, and energy-efficient spiking neural networks?[C]//ICASSP 2022-2022 IEEE International Conference on Acoustics, Speech and Signal Processing (ICASSP). IEEE, 2022: 71-75.

[2] Han B, Roy K. Deep spiking neural network: Energy efficiency through time based coding[C]//European Conference on Computer Vision. Cham: Springer International Publishing, 2020: 388-404.

[3] Comsa I M, Potempa K, Versari L, et al. Temporal coding in spiking neural networks with alpha synaptic function[C]//ICASSP 2020-2020 IEEE International Conference on Acoustics, Speech and Signal Processing (ICASSP). IEEE, 2020: 8529-8533.

**Questions:**

1. I suppose the traditional positional encoding in Spikformer can be seen as direct encoding in SNNs. In contrast, the CPG-PE proposed in this paper is more like a temporal encoding with dynamic expressions related to time series. Could the authors discuss the differences between other encoding schemes [1;2;3] and positional encoding in SNNs within the related work?

2. Can the authors provide experimental results of CPG-PE on the ImageNet image classification dataset?
3. Some tyops: Formulas 4, 5, 6, 7, 8, and 10 are missing commas. There should be a comma after each formula.

I would be pleased to raise the score if the authors address my concerns.

[1] Kim Y, Park H, Moitra A, et al. Rate coding or direct coding: Which one is better for accurate, robust, and energy-efficient spiking neural networks?[C]//ICASSP 2022-2022 IEEE International Conference on Acoustics, Speech and Signal Processing (ICASSP). IEEE, 2022: 71-75.

[2] Han B, Roy K. Deep spiking neural network: Energy efficiency through time based coding[C]//European Conference on Computer Vision. Cham: Springer International Publishing, 2020: 388-404.

[3] Comsa I M, Potempa K, Versari L, et al. Temporal coding in spiking neural networks with alpha synaptic function[C]//ICASSP 2020-2020 IEEE International Conference on Acoustics, Speech and Signal Processing (ICASSP). IEEE, 2020: 8529-8533.

**Limitations:**

The author has discussed the limitations

---

> ### Author Rebuttal · Authors · 2024-08-06
>
> We are cheerful that our contribution is well recognized. And thanks for your valuable suggestions that truly enhance the quality of our paper and make it more understandable for the wider community.
>
> Responses to your concerns are presented as follows:
>
> ### **1.ImageNet Dataset (W1, Q2)**
>
> Thanks for your suggestion. We have conducted experiments with Spikformer without positional encoding (PE), Spikformer with relative positional encoding (RPE), and Spikformer with our proposed CPG-PE on the ImageNet dataset. The results are as follows:
>
> | Model                       | Param(M) | ImageNet |
> | --------------------------- | -------- | -------- |
> | Spikformer w/o PE           |    $15.50$   |   $69.46$%    |
> | Spikformer w/ RPE \[1\]      |    $16.81$     |   $70.24$%    |
> | Spikformer w/ CPG-PE \[ours\] |    $15.66$   |    $71.17$%   |
>
> Specifically, we set the depth to 8 and the dimension of representation to 384. For other experimental settings, we have faithfully followed the guidelines in \[1\]. From the table, we can see that CPG-PE performs well on large-scale image datasets. However, due to time constraints, we were unable to conduct experiments on Random PE or Float PE. We believe that the above results demonstrate the effectiveness of our proposed CPG-PE in positional encoding. We will add these experiment results to Table 3 in our revised manuscript.
>
>
> ### **2.Discussion on temporal coding and rate coding (W2, Q1)**
>
>  Thanks for your suggestion.
>
> (1) We carefully reviewed the literature you provided and found your suggestions valuable for improving the quality of our paper. In the revised manuscript, we will include the following discussion in the related work section (due to character limitation, we omit some references but will include them in the revised manuscript):
>
> "Spiking Neural Networks (SNNs) employ several coding methods to encode input information, each offering unique advantages. Direct coding \[1\]\[2\], the simplest form and widely-used in image tasks, directly associates spikes with specific values or events, providing straightforward and interpretable outputs but often lacking efficiency for complex tasks. Rate coding \[3\]\[4\], where the input is represented by the frequency of spikes within a given timeframe, is more robust and widely used but can be less precise due to its reliance on averaged spike rates. Temporal coding \[5\]\[6\] (a.k.a latency coding) encodes information based on the timing of individual spikes, allowing for high temporal precision and efficient representation of dynamic inputs, though it can be computationally demanding. In addition, Delta coding \[7\] represents changes in input signals through spikes, focusing on differences rather than absolute values, which can enhance efficiency and response times but may introduce complexity in decoding. Each of these methods contributes to the versatility and applicability of SNNs in various domains, from neuroscience to artificial intelligence. The SNNs we considered in this paper should fall into the category of rate coding since back-prop is conducted on spike rate. Meanwhile, CPG-PE can be considered converting temporal information into spike rate of a group of neurons (Equations 11-12), and this is why CPG-PE can improve performance for sequential data. It is possible to introducing learning algorithms of temporal coding for the CPG neurons to tackle more complex sequence structure, which remains as future work."
>
> (2) In Related Works Section, we have discussed the positional encoding used in SNNs.
>
> ###  **3.Typo (Q3)**
>  Thanks for your reminding. We will add commas after formulas 4-10 to ensure readers are not misled.
>
> ### Reference
>
> [1] Zhou Z, Zhu Y, He C, et al. Spikformer: When Spiking Neural Network Meets Transformer[C]//The Eleventh International Conference on Learning Representations (ICLR). 2023.
>
> [2] Yao M, Hu J, Zhou Z, et al. Spike-driven transformer[J]. Advances in Neural Information Processing Systems (NeurIPS), 2023.
>
> [3] Kim Y, Park H, Moitra A, et al. Rate coding or direct coding: Which one is better for accurate, robust, and energy-efficient spiking neural networks?[C]//ICASSP 2022-2022 IEEE International Conference on Acoustics, Speech and Signal Processing (ICASSP). IEEE, 2022: 71-75.
>
> [4] Lv C, Xu J, Zheng X. Spiking Convolutional Neural Networks for Text Classification[C]. // The Eleventh International Conference on Learning Representations (ICLR). 2023.
>
> [5] Han B, Roy K. Deep spiking neural network: Energy efficiency through time based coding[C]//European Conference on Computer Vision. Cham: Springer International Publishing, 2020: 388-404.
>
> [6] Comsa I M, Potempa K, Versari L, et al. Temporal coding in spiking neural networks with alpha synaptic function[C]//ICASSP 2020-2020 IEEE International Conference on Acoustics, Speech and Signal Processing (ICASSP). IEEE, 2020: 8529-8533.
>
> [7] Yoon Y C. LIF and simplified SRM neurons encode signals into spikes via a form of asynchronous pulse sigma–delta modulation[J]. IEEE transactions on neural networks and learning systems, 2016, 28(5): 1192-1205.

---

> > ### Comment · Reviewer_Pjez · 2024-08-12
> >
> > Thank you for your reply. I think this is a nice bit of discussion and could be added to the manuscript. In light of the additional discussion, I'd like to raise my score to a 7. This is an interesting piece of work and would be a nice addition to NeurIPS.

---

### Official Review · Reviewer_s3xa · 2024-07-11

**Soundness:** 3
**Presentation:** 3
**Contribution:** 3
**Rating:** 6
**Confidence:** 5

**Summary:**

This paper introduces a novel positional encoding method for SNNs called CPG-PE, inspired by central pattern generators (CPGs) in biological neural systems. The authors demonstrate both theoretically and empirically that CPG-PE can effectively capture positional information in sequential data while maintaining the spike-based nature of SNNs. The approach is evaluated on a range of tasks including time-series forecasting, text classification, and image classification.

**Strengths:**

1. Strong theoretical foundation: The authors mathematically demonstrate how CPG-PE relates to conventional sinusoidal positional encoding used in transformers.
2. Comprehensive empirical evaluation across time-series forecasting, text classification, and image classification tasks shows consistent performance improvements when incorporating CPG-PE.
3. The method is biologically plausible and potentially compatible with neuromorphic hardware, as it can be implemented using leaky integrate-and-fire neurons.
4. Well-structured paper with clear motivation, methodology, and insightful analysis of CPG-PE properties and their relationship to biological CPGs.

**Weaknesses:**

1. Typographical error on line 158 where X is incorrectly specified as belonging to the real number domain instead of the binary domain {0,1} for spike data.
2. The authors should clarify the similarities and differences between the decay mechanism in CPG-PE and that of LIF neurons.

**Questions:**

See my weakness part.

**Limitations:**

The authors have addressed limitations and future work in Appendix E, which is commendable. They discuss the challenges of applying CPG-PE to non-sequential data like images and propose potential solutions.

---

> ### Author Rebuttal · Authors · 2024-08-06
>
> We appreciate your comments and suggestions that enhanced the quality of our paper. Responses to your concerns and questions are hereby presented:
>
> ###  **1.Typographical error (W1)**
>
> Apologies for the confusion caused by this typo error. In line 158, the $X$ is the input spike matrix and it belongs to  the binary domain $\{0,1\}$ rather than real number domain $\mathbb{R}$, i.e., $X \in \{0,1\}^{T \times B \times L \times D}$. This error will be corrected in our revised manuscript.
>
> ### **2.Similarities and differences between the decay mechanism of CPG-PE and that of LIF neurons (W2)**
>
> (1) If we understand your question correctly (please kindly let us know if we misunderstood), the question is about the differences between the decay mechanisms of *CPG neurons* and LIF neurons instead of *CPG-PE* . To be clear, CPG indicates that the neuron (together with other neurons in a CPG circuit) plays the role of rhythmic output generator, while LIF is a specific dynamics of spiking neuron model. CPG-PE is our propose method which leverages the spikes of CPG neurons for encoding positional information in sequential data. Therefore, CPG-PE and LIF neurons are orthogonal concepts, and we have proved that CPG-PE could be composed of LIF neurons (as shown in Appendix C), or be manually set (for simplicity, as used in our experiments).
>
> (2) Regarding the decay mechanism, as shown in Equation (2), the $\beta$ control the decay rate of LIF neurons. However, our proposed CPG-PE does not have a decay mechanism. We would greatly appreciate it if you could provide further clarification on this issue, and we are happy to address it.
>
> Please let us know if you have any remaining questions or concerns, we are commited to addressing the issues.

---

> > ### Comment · Reviewer_s3xa · 2024-08-13
> > **Thank you**
> >
> > The reviewer thanks the authors for the discussion. It addressed all of my concerns. I decide to keep my score.

---

### Official Review · Reviewer_3Lmi · 2024-07-12

**Soundness:** 3
**Presentation:** 3
**Contribution:** 3
**Rating:** 7
**Confidence:** 5

**Summary:**

The lack of an effective and hardware-efficient spike-frm position encoding strategy in  SNNs has been a consistent motivation for this study.
Drawing inspiration from the central pattern generators (CPGs) in the human brain, which produce rhythmic patterned outputs without requiring rhythmic inputs, this work proposes a novel PE technique for SNNs, termed CPG-PE.
Extensive experiments across various domains show the superior performance with CPG-PE.

**Strengths:**

1.To the best of my knowledge, this is the first work on position encoding in SNNs, laying the foundation for efficient sequence modeling in SNNs.

2.The approach utilizes the coupling of multi-neuron pulse signals as position encoding, which is innovative.

3.The authors demonstrated the effectiveness of this method across various tasks.

4.The spike-position encoding generating method through mutual inhibition between two groups of spiking neurons is brain-inspired and hardware-friendly.

**Weaknesses:**

1.This type of positional encoding, through aggregation, introduces a small number of additional parameters and computational overhead. Please provide ablation experiments demonstrating that the performance improvement is not solely due to these factors.
2.I suggest the authors include results on ImageNet to demonstrate the effectiveness on large-scale datasets.

**Questions:**

1.In ANNs, positional encoding is typically added to features. Please analyze the similarities and differences of this category aggregation-based positional encoding compared to ANNs.

**Limitations:**

I believe the authors' discussion on limitations is comprehensive.

---

> ### Author Rebuttal · Authors · 2024-08-06
>
> Thank you for your comments and suggestions, which are valuable for enhancing our paper. We are pleased that our contributions are well recognized.
>
> Responses to your concerns and questions are hereby presented:
>
> ###  **1.Ablation experiments on parameters (W1).**
>
> Thanks for your suggestion.
>
> (1) Firstly, while our method increases the parameter count compared to the version without positional encoding (w/o PE), this increase is relatively small. For example, on CIFAR10, the paramter numbers of Spikformer with CPG-PE are only $0.17$M(~$2.17$% of the total parameters) larger than that of Spikformer w/o PE.
>
> (2) Secondly, we conducted ablation experiments by reducing the parameter count of Spikformer with CPG-PE to be comparable to Spikformer w/o PE, allowing for a more direct performance comparison. The results are as follows:：
>
> | Model                               | CIFAR10                        | CIFAR10-DVS                    | CIFAR100                       |
> | ----------------------------------- | ------------------------------ | ------------------------------ | ------------------------------ |
> | Spikformer w/o PE                   | Param: $8.00$M, Accuracy: $93.77$% | Param: $1.99$M, Accuracy: $76.40$% | Param: $8.04$M, Accuracy: $73.59$% |
> | Spikformer w/ CPG-PE \[Equal Param\] | Param: $7.99$M, Accuracy: $94.60$%     | Param: $1.99$M, Accuracy: $78.00$% | Param: $8.02$M, Accuracy: $76.91$%     |
>
> Specifically, we adjusted the representation dimension of Spikformer with CPG-PE to make the parameter counts similar to those of Spikformer without PE. For instance, on CIFAR10, we changed the dimension from $384$ to $380$, resulting in a parameter count of $7.99$M, which is almost equal to $8.00$M. As shown in the table above, we can conclude that when the parameter counts are equal, Spikformer with CPG-PE significantly outperforms Spikformer without PE across all image classification datasets, proving our proposed CPG-PE is effective.
>
> ### **2.ImageNet Dataset (W2).**
>
> Thanks for your suggestion. We have conducted experiments with Spikformer without positional encoding (PE), Spikformer with relative positional encoding (RPE), and Spikformer with our proposed CPG-PE on the ImageNet dataset. The results are as follows:
>
> | Model                       | Param(M) | ImageNet |
> | --------------------------- | -------- | -------- |
> | Spikformer w/o PE           |    $15.50$   |   $69.46$%    |
> | Spikformer w/ RPE \[1\]      |    $16.81$     |   $70.24$%    |
> | Spikformer w/ CPG-PE \[ours\] |    $15.66$   |    $71.17$%   |
>
> Specifically, we set the depth to 8 and the dimension of representation to 384. From the table, we can see that CPG-PE performs well on large-scale image datasets. However, due to time constraints, we were unable to conduct experiments on Random PE or Float PE. We believe that the above results demonstrate the effectiveness of our proposed CPG-PE in positional encoding. We will add these experiment results to Table 3 in our revised manuscript.
>
> ### **3.Similarities and differences between CPG-PE and addition-based PE in ANNs (Q1).**
>
> In ANNs, especially in language tasks, the difference between using addition and concatenation for positional encoding is not significant \[2\]\[3\]. However, in spiking neural networks (SNNs), addition is not suitable for spike matrices because it can result in non-binary values, which is one of the motivations for designing SNN-PE. This paper addresses this issue by first concatenating the spike matrix and then using a projection layer to project it back to the initial dimension.
>
> ### Reference
>
> [1] Zhou Z, Zhu Y, He C, et al. Spikformer: When Spiking Neural Network Meets Transformer[C]//The Eleventh International Conference on Learning Representations (ICLR). 2023.
>
> [2] Rosendahl J, Tran V A K, Wang W, et al. Analysis of positional encodings for neural machine translation[C]//Proceedings of the 16th International Conference on Spoken Language Translation. Association for Computational Linguistics. 2019.
>
> [3] Ke G, He D, Liu T Y. Rethinking Positional Encoding in Language Pre-training[C]//International Conference on Learning Representations (ICLR). 2021.

---

> ### Comment · Reviewer_3Lmi · 2024-08-10
> **Response  to the rebuttal**
>
> Thank you for your detailed response. The experiments have alleviated my concerns about the overhead of CPG-PE. The further validation on the large-scale ImageNet has demonstrated its generalization and effectiveness. Therefore, I would like to increase my score.

---

### Official Review · Reviewer_P6YU · 2024-07-12

**Soundness:** 2
**Presentation:** 2
**Contribution:** 2
**Rating:** 6
**Confidence:** 4

**Summary:**

This paper introduces central pattern generators (CPGs) from neuroscience into the SNN framework as a novel method for position encoding. Through mathematical derivation, it is proven that the existing abstract PE methods in transformers are actually a particular solution for a specific type of CPG. The effectiveness of CPG is validated through experiments across several domain benchmarks.

**Strengths:**

1.	This article connects existing abstract Positional Encoding (PE) methods in transformers with Central Pattern Generators (CPGs) in the human brain through mathematical derivation, showing that the former can be viewed as a specific mathematical solution to the membrane potential dynamics of the latter. This presents an interesting viewpoint.

2.	The paper is well-organized and clearly written, offering high readability. Readers can effortlessly grasp the authors' intentions, supported by both the textual explanations and accompanying illustrations.

**Weaknesses:**

1.	The authors have not convinced me why the problem addressed in this paper is very important, i.e., why the existing PE methods in SNNs are such a big problem that they need to be improved by CPG.


2.	The method proposed in the paper is simple and does not provide enough inspiring insights; the contribution is relatively limited.

3.	The implementation of CPG-PE on hardware involves coupled nonlinear oscillators that require frequent updates of neuron membrane potentials, entailing floating-point computations and memory read-write operations, which result in additional energy expenditures. The paper should scrutinize and analyze whether the performance enhancements afforded by this encoding method justify the additional energy costs. This trade-off demands a detailed examination to assess its viability in practical applications.

4.	In Sec3.1, why is "F(x)=b<=0, H(y)=d<=0" followed after “...gain membrane voltage with constant speed” instead of “F(x)=b>0, H(y)=d>0”?

5.	In Sec4.2, does the CPG-Full method replace all linear layers in the model with CPG-Linear layers? Why is its performance not as good as that of CPG-PE? Can you provide further analysis and explanation?

**Questions:**

Please refer to weakness 3, 4, 5

---

> ### Author Rebuttal · Authors · 2024-08-06
>
> Thanks for your comments and suggestions, which are valuable for enhancing our paper. The follows are responses to your individual concerns:
>
> ### **1.Why is positional encoding important to SNNs (W1)?**
> Currently, SNNs have been applied to a variety of tasks beyond image processing, including time-series forecasting [1] and text classification [2]. However, due to lack of suitable positional encoding (PE) method, it remains challenging for SNNs to capture indexing information, rhythmic patterns, and periodic data. Existing PE methods for SNNs do not meet the essential criteria: uniqueness of each position and formulation in spike-form. Inspired by CPG neurons, we propose CPG-PE, which performs well on both sequential and image tasks. Additionally, we ensure its hardware-friendliness and provide insights into the role of CPGs in neuroscience (Section 6).
>
> ### **2.The method is simple and does not provide enough inspiring insights; contribution is relatively limited. (W2).**
> > "Everything should be made as simple as possible, but no simpler." – Albert Einstein
>
> (1) We believe simplicity is a strength of our method (as it is effective), not a weakness. As it is easy to implement, there is a higher probability that more researchers can benefit from leveraging our method on their model to improve performance. Furthermore, since CPG-PE is novel to the SNN community, complex models and algorithms with even better capacity may emerge following our work, but it is necessary that we first comprehensively address that a simple CPG-PE can consistently improve the performance of SNNs.
>
> (2) Our contributions can be summarized as: **(1) Pioneer in the exploration of spike-version PE for SNNs.** To the best of our knowledge, our proposed CPG-PE is the first work on spike-version PE in SNNs, which are both brain-inspired and hardware-friendly. We corrected the inappropriate PE designs previously used in SNNs. **(2) Consistent Performance Gain.** CPG-PE enhances the performance of SNNs across a wide range of tasks, including time-series forecasting, text classification, and image classification. **(3) Insights on neuroscience and neuromorphic hardware.** We mathematically proved that the traditional PE in Transformers is a particular solution of the membrane potential variations in a specific type of CPG, and we also proved that CPG-PE can be implemented with 2 LIF neurons so that it will not introduce any burden of redesigning hardware.
>
> ### **3.Frequent updates of neuron membrane potentials bring additional floating-piont operations and energy on hardware (W3).**
> We fully agree, and indeed, we have discussed this aspect in our paper.
>
> (1) **Physical Implementation of CPG-PE** We have proposed an approach to physically implement CPG-PE with only 2 LIF neurons, introducing no extra efforts on chip designs.
> A CPG-PE neuron can be viewed as an autonomic neuron that will emit a burst of $K$ spikes after resting for $R$ time steps. Through rigorous derivation (see **Appendix C**), we prove that the system can behave periodically with a period $T = (R + K)∆T$, i.e., CPG-PE can be implemented with 2 LIF neurons. This approach eliminates the need for additional floating-point computations and memory read-write operations on neuromorphic hardware to frequently update neuron membrane potentials.
>
> (2) **Support from Existing Circuit Implementations** Existing circuit implementations on hardware [3][4] strongly support our proposed CPG-PE, as these works are based on the similar idea of **coupled LIF neurons**. Circuits based on complementary metal-oxide-semiconductor technology [3] simplify membrane potential calculations, reduce power consumption significantly, and eliminate the need for additional memory to store membrane potentials. Furthermore, memristor-based CPG circuits [4] have been shown to minimize circuit complexity and increase energy efficiency.
> Therefore, we believe that our proposed approach is hardware-friendly, without requiring additional operations on hardware, which has also been acknowledged by other reviewers (3Lmi, s3xa, Pjez).
>
> ### **4.Misleading in Section 3.1 (W4).**
> We apologize for this typo and the resulting confusion. It should be: "$\mathbf{F}(\mathbf{x})=b>0, \mathbf{H}(\mathbf{y})=d>0$."
> After our thorough inspection and verification, we confirm that this does not affect the derivation results, and Equations 6-10 remain valid. We will correct this typo in our revised manuscript.
>
> ###  **5.Why is the performance of CPG-Full not as good as that of CPG-PE (W5).**
>
> (1) CPG-Full stands for replacing all linear layers with CPG-Linear layers. CPG-Linear was introduced in Appendix D. The original CPG-PE models the inputs, while CPG-Full models the hidden states.
>
> (2) The difference between the two is not significant. In terms of metrics, the average $R^2$ is identical, and the $RSE$ difference is not statistically significant (p-value=$0.9351$, student *t*-test).
>
> (3) We conducted this experiment for two main reasons: Firstly, to explore how to implement modular CPG, hypothesizing that simply replacing Linear with CPG-Linear would allow the PE functionality to work effectively. Secondly, this experiment is supplementary, aiming to confirm that the performance improvement of SNNs is due to the functionality of PE rather than an enhancement in representation capability, as CPG-Full does not affect representation capability.
>
> Please kindly let us know if there is any remaining concern. We are committed to addressing your concerns.
>
> [1] Lv C, et al. Efficient and Effective Time-Series Forecasting with Spiking Neural Networks. ICML, 2024.
>
> [2] Bal M, et al. Spikingbert: Distilling bert to train spiking language models using implicit differentiation. AAAI, 2024.
>
> [3] Vogelstein J, et al. Dynamic control of the central pattern generator for locomotion. Biological Cybernetics, 2006.
>
> [4] Dutta S, et al. Programmable coupled oscillators for synchronized locomotion. Nature Communications, 2019.

---

> > ### Comment · Reviewer_P6YU · 2024-08-09
> > **Response to the rebuttal**
> >
> > Thank you for the response to my questions. The idea of building positional encoding in SNN is interesting and I am willing to increase my score to 6.

---

### Author Rebuttal · Authors · 2024-08-06

# Global Response

We express our gratitude to all the reviewers for the valuable insights and acknowledging our contributions to advance the sequential modeling ability of SNNs through central pattern generators. We are encouraged by the comments highlighting the strengths of our work:

- Clear motivation, novelty, and innovation (Reviewer 3Lmi, s3xa)
- Effectiveness / consistent performance improve (Reviewer 3Lmi, s3xa, Pjez)
- Both Hardware-friendliness and biologically-plausibility (Reviewer 3Lmi, s3xa, Pjez)
- Comprehensive experiments and analysis (Reviewer 3Lmi, s3xa, Pjez)
- Well-organized and clearly-written (Reviewer P6YU, s3xa)

A major concern most reviewers share is the evaluation on large-scale image classification benchmark, i.e., ImageNet. We agree with the comment and to address this issue, we have conducted experiments with Spikformer without positional encoding (PE), Spikformer with relative positional encoding (RPE), and Spikformer with our proposed CPG-PE on the ImageNet dataset. The results are as follows:

| Model                       | Param(M) | ImageNet Acc (Top 1)|
| --------------------------- | -------- | -------- |
| Spikformer w/o PE           |    $15.50$   |   $69.46$%    |
| Spikformer w/ RPE       |    $16.81$     |   $70.24$%    |
| Spikformer w/ CPG-PE \[ours\] |    $15.66$   |    $71.17$%   |

In particular, we set the depth to 8 and the dimension of representation to 384. From the table, we can see that CPG-PE performs well on large-scale image datasets. However, due to time constraints, we were unable to conduct experiments on Random PE or Float PE. We believe that the above results demonstrate the effectiveness of our proposed CPG-PE in positional encoding.

One comment on weakness is about our contributions. We would like to highlight several key contributions that distinguish our work from existing literature: **(1) Pioneer in the exploration of positional encoding for SNNs.** To the best of our knowledge, our proposed CPG-PE is the first work on spike-version positional encoding in SNNs, which are both brain-inspired and hardware-friendly. We corrected the inappropriate positional encoding designs previously used in SNNs. **(2) Consistent performance gain with a simple method.** CPG-PE enhances the performance of SNNs across a wide range of tasks, including time-series forecasting, text classification, and image classification. **(3) Insights on neuroscience and neuromorphic hardware.** We mathematically proved that the traditional positional encoding in Transformers is a particular solution of the membrane potential variations in a specific type of CPG. Additionally, we demonstrated that CPG-PE can be implemented with 2 LIF neurons, ensuring it does not introduce any burden of redesigning hardware.


We will address the concerns and polish our paper in the revised version to enhance its clarity and accessibility for the wider community. To summarize the updates:

1. Extensive experiment results on large-scale dataset, specifically ImageNet.
2. Further discussion on spike encoding methods, including temporal coding and rate coding, in the related work section.
3. Revision of typos and proofreading for better language.

We are confident that our work contributes to the NeurIPS community by advancing neuromorphic AI and potentially computational neuroscience. We are happy to answer follow-up questions from the reviewers if anything remains unclear.

---

### Decision · Program_Chairs · 2024-09-25

**Decision:**

Accept (spotlight)

**Comment:**

All reviewers agree that this is an interesting work that can be accepted for NeurIPS. The paper proposes biologically inspired central pattern generators (CPGs) as a novel method for position encoding in spiking neural networks (SNNs).
The reviewers noted that the work
-    covers new ground for sequential processing with SNNs,
-    has a strong theoretical foundation,
-    provides a comprehensive empirical evaluation on a variety (and also large) data sets,
-    shows significantly improved performance over previous SNN models,
-    is well written and has insightful analysis.
I therefore recommend to accept the paper.